# Dual Mechanism of Action of Curcumin in Experimental Models of Multiple Sclerosis

**DOI:** 10.3390/ijms23158658

**Published:** 2022-08-04

**Authors:** Ines ELBini-Dhouib, Maroua Manai, Nour-elhouda Neili, Soumaya Marzouki, Ghada Sahraoui, Warda Ben Achour, Sondes Zouaghi, Melika BenAhmed, Raoudha Doghri, Najet Srairi-Abid

**Affiliations:** 1Laboratoire des Biomolécules, Venins et Applications Théranostiques (LR20IPT01), Institut Pasteur de Tunis, Université de Tunis El Manar, Tunis 1002, Tunisia; 2Laboratoire de Génétique Humaine (LR99ES10), Faculté de Médecine de Tunis, Université de Tunis El Manar, Tunis 2092, Tunisia or; 3Laboratoire de Transmission, Contrôle et Immunobiologie des Infections (LR11IPT02), Institut Pasteur de Tunis, Université de Tunis El Manar, Tunis 1002, Tunisia; 4Laboratoire de Médecine de Précision, Médecine Personnalisée et Investigation en Oncologie (LR21SP01), Service d’Anatomie Pathologique, Institut Salah Azaiez, Bab Saadoun, Tunis 1006, Tunisia; 5Faculté de Médecine de Tunis, Université de Tunis El Manar, Tunis 1068, Tunisia

**Keywords:** neuroinflammation, demyelination, curcumin

## Abstract

Background: Multiple sclerosis (MS) is characterized by a combination of inflammatory and demyelination processes in the spinal cord and brain. Conventional drugs generally target the autoimmune response, without any curative effect. For that reason, there is a great interest in identifying novel agents with anti-inflammatory and myelinating effects, to counter the inflammation and cell death distinctive of the disease. Methods and results: An in vitro assay showed that curcumin (Cur) at 10 µM enhanced the proliferation of C8-D1A cells and modulated the production of Th1/Th2/Th17 cytokines in the cells stimulated by LPS. Furthermore, two in vivo pathophysiological experimental models were used to assess the effect of curcumin (100 mg/kg). The cuprizone model mimics the de/re-myelination aspect in MS, and the experimental autoimmune encephalomyelitis model (EAE) reflects immune-mediated events. We found that Cur alleviated the neurological symptomatology in EAE and modulated the expression of lymphocytes CD3 and CD4 in the spinal cord. Interestingly, Cur restored motor and behavioral deficiencies, as well as myelination, in demyelinated mice, as indicated by the higher index of luxol fast blue (LFB) and the myelin basic protein (MBP) intensity in the corpus callosum. Conclusions: Curcumin is a potential therapeutic agent that can diminish the MS neuroimmune imbalance and demyelination through its anti-inflammatory and antioxidant effects.

## 1. Introduction

Multiple sclerosis (MS) is a disease commonly characterized by inflammation and demyelination in the central nervous system (CNS) [1], the onset of which predominantly occurs between the ages of 20 and 40 years, with a greater prevalence in females [2]. The clinical manifestations of MS are highly variable, making the diagnosis of the disease difficult [3]. Imagery investigation shows multifocal inflamed regions in gray and white matter, with oligodendrocyte death and myelin sheath disruption [4]. In the initial phase of the disease, myelin-reactive T cells, as well as immunocytes such as macrophages in the periphery, cross the disrupted blood–brain barrier (BBB) and infiltrate the CNS [5]. The inflammatory response is mainly mediated by T helper (Th) cells [5]. Then, activated CD4+ T cells produce a substantial number of pro-inflammatory factors, which results in demyelination and eventual axonal degeneration [6]. Besides this, activation of CNS resident cells, particularly microglia and astrocytes, also increases the pathogenesis of MS [7]. Microglia cells are activated when tissue integrity is disturbed, secreting pro-inflammatory or anti-inflammatory cytokines, controlling the expression of anti-inflammatory molecules, regulating phagocytosis of debris and supporting tissue repair [8]. Most importantly, astrocytes strengthen and contribute to the maintenance of the integrity of the BBB, restricting the entry of peripheral immune cells into the CNS and modulating synaptic activity and plasticity [9]. Astrocytes also promote the migration, proliferation and differentiation of oligodendrocyte precursor cells, and they are responsible for the production of mediators of inflammation [10], as well as various factors that stimulate or inhibit myelination in MS [11,12].

On the other hand, the early phase of MS is also associated with demyelination of axons, which is recovered with new myelin sheaths in a spontaneous regenerative process, called remyelination. This process is mediated by resident oligodendrocyte progenitor cells (OPC) that differentiate into new myelinating oligodendrocytes [13]. As the disease progresses, this process declines, partially due to impaired OPCs’ generation or failure of their differentiation to mature myelinating oligodendrocytes [14,15].

Currently, several immunosuppressive and immunomodulatory therapies are broadly used to modulate immune responses via several mechanisms. However, these therapeutic approaches have limited curative outcomes and are associated with side effects. Therefore, it is necessary to investigate the effects of natural compounds on the pathogenesis processes of the disease in preclinical models. Fortunately, experimental models of MS consisting of experimental autoimmune encephalomyelitis (EAE) and cuprizone models provide an excellent tool for screening potential anti-MS compounds [16,17,18]. EAE has an auto-inflammatory nature, where the cuprizone model is widely used to study the demyelination aspect of the disease [19]. Among the complementary approaches, chemical and synthetic drugs are gradually being replaced by more natural alternatives with fewer to no side effects. In this context, curcumin (Cur), from *Curcuma longa* [20], was reported to have multiple advantageous effects, including antioxidant [21] and anti-inflammatory ones [22]. In addition, the efficiency of Cur against several inflammatory diseases such as neurodegenerative diseases [23,24] and cancer [25] has been highlighted in previous studies. In the present work, we investigated the potential therapeutic effect of curcumin on autoimmune (EAE) and toxic (cuprizone) animal models of MS, as well as on an inflamed astrocyte cell line culture.

## 2. Results

### 2.1. Curcumin Is Not Cytotoxic and Promotes C8-D1A Proliferation

The effect of curcumin was first assessed on the viability of C8-D1A cells at different concentrations (10 μM, 20 μM, and 50 μM) via MTT assay. Data showed that curcumin at a high dose (50 μM) induced cell death in astrocyte cells (*** *p* < 0.001) (Figure 1A), whereas it did not affect cell viability at 10 μM compared to the non-treated group (NT) (* *p* < 0.05) (Figure 1A). To test whether curcumin influences the proliferation of the astrocyte cell line, depicted by the length of cell connections, a cell count was performed at different time points during the treatment (24, 36 and 72 h) and photomicrographs were taken (Leica Software V4.13). After 24 h, cell proliferation of Cur-treated astrocytes significantly increased compared to NT cells with normal growth behavior (Figure 1B). The effect of curcumin on cell proliferation was dose-dependant. On the other hand, NT cells exhibited a constant and nearly linear growth (100% growth). After 72 h of treatment, the count of C8-D1A cells treated with 10 μM (CUR1) increased by an average of 19.33% (*p* < 0.01) (Figure 1B). However, morphological assessment of cells treated with 50 µM of Cur (CUR3) showed cell damage such as cell loss, shorter cell processes and cell shrinkage in comparison to the NT and CUR1 conditions (Figure 1B). Therefore, curcumin at high concentrations induced cytotoxicity and cell damage (Figure 1B,C).

### 2.2. Curcumin Immunomodulates Inflammatory C8-D1A Induced by LPS

Th1/Th2/Th17 cytokines were measured in the cell culture of C8-D1A after 2 h of pre-incubation with curcumin at 10 µM followed by 18 h of additional LPS (10 mg/mL). TNFα, IL-2, IL-6, IFNγ and IL-17A cytokines release increased significantly in LPS-stimulated cells compared to non-stimulated cells (Figure 2). Cur at 10 µM significantly decreased the release of TNFα, IL-2, IL-6, IFNγ and IL-17A cytokines by 52.36, 31.8, 66.6, 58.4 and 42.5%, respectively, whereas it significantly increased IL-10 release in LPS-stimulated cells (*** *p* < 0.001). 

### 2.3. Curcumin Ameliorates the Neurological Severity of EAE Mice

While no neurological changes were observed in the negative control group (CTR-), EAE mice showed signs of decreased activity and nutritional behavior after immunization. The first EAE signs appeared 12 days following the immunization of mice with MOG_35–55_/CFA and PTX (Figure 3A). The maximal degree of paralysis, indicating an acute phase, occurred around day 21 when the average neurological score reached 4.29, up until day 30. From this point forward, animals treated with Cur partially recovered from paralysis with a decrease in the neurological score, and the disease stabilized around day 45 with an average score of 1.25 (Figure 3B), while the neurological score remained at 4.05 in spontaneous-recovery mice (Srec). In addition, body weight decreased in EAE mice when compared to the CTR- group, whereas a slight weight recovery was observed in animals treated with curcumin (Figure 3B). 

### 2.4. Curcumin Decreases Inflammatory Cells’ Infiltration and Demyelination in Spinal Cord of EAE Mouse 

To further study the effect of curcumin on the pathogenesis of CNS inflammation, we investigated its effect on the spinal cord of EAE animal model. In this study, animals were sacrificed before or following the curcumin treatment period. Mice from the CTR- and EAE groups were sacrificed on day 30 (at the peak of the disease). At 45 days, animals from the curcumin-treated (100 mg/kg of curcumin) and spontaneous-recovery groups were also sacrificed. Spinal cord sections from all groups of mice were stained with LFB and hematoxylin and eosin (HE) to assess demyelination and inflammation, respectively. As shown in Figure 4, the EAE mice showed profound inflammatory cell infiltration (Figure 4A,B) and demyelination (Figure 4C) in the spinal cord; conversely, treatment with curcumin significantly decreased inflammatory cell infiltration and demyelination. It is worth mentioning that pathological changes due to the EAE model persisted in the spontaneous recovery group; it was thus used as a positive control group in parallel with the curcumin-treated group (Figure 4A–C).

To identify the types of immune cells that infiltrate spinal cord tissue of the EAE model, we carried out an immunohistochemistry analysis to highlight the expression levels of LTCD3, LTCD4 and LTCD8. The expression of LTCD3 and LTCD4 cells in the spinal cord was significantly increased in the EAE and Srec groups, while they decreased in the group treated with curcumin when compared with the EAE and Srec groups (Figure 5).

### 2.5. Curcumin Improves the Behavioral Abnormalities Induced by Cuprizone Administration in Mice

Cuprizone was administered via gavage for 8 weeks in female mice as the first step of this study, followed by 15 days of curcumin treatment as a second step. The body weight of the mice was measured daily throughout the experimental period (first and second steps). After 5 days of cuprizone administration, the treated animals had lost 2.7% of their body weight, while animals on a normal diet had gained 8.01% (data not presented). Over the next several days, treated animals regained the same weight as the control group. After 8 weeks of cuprizone administration, the differences between the control and CPZ mice were statistically significant (*p* < 0.001) (Figure 6A). CPZ animals had an overall decrease of 23.5% in their body weight with respect to the initial values, while control animals achieved a 50.9% increase (*p* < 0.001) (Figure 6A). It is worth noting that no deaths occurred during the experimental procedure and post-treatment, with curcumin improving the body weights of mice when compared to the SR group. 

In our behavioral studies, we assessed the memory, anxiety and motricity of mice in all experimental groups.

Initially, we did not notice a significant difference in the frequency of sniffing between the familiar objects (Familiar 1 versus Familiar 2) among the groups during the familiarization phase of the NORT. During the test phase, the frequency of sniffing of the novel object was significantly higher than the familiar object in the control (*p* < 0.001) and CUR groups (*p* < 0.001), but there was no difference in the CPZ or SR groups (Figure 6B). 

Then, the CPZ group exhibited a significant decrease in the Discrimination Index (DI) compared to the control group (*p* < 0.001), while the CUR group showed a significant increase in the DI compared to the CPZ group (*p* < 0.001) (Figure 6B). The pole test showed that cuprizone increased the time spent on touchdown (*p* < 0.01), which was significantly shortened by curcumin treatment compared to the CPZ and SR groups (*p* < 0.05) (Figure 6C). 

The explorative behavior of mice was tested in a light/dark box analysis. We found that CPZ mice spent 50% less time in the lit compartment when compared to the CTR- group (Figure 6D; *p* < 0.01). 

Furthermore, Kondziela’s inverted screen test responses were significantly affected by cuprizone (*p* < 0.001), with scores significantly higher in CUR mice compared to CPZ mice (Figure 6E).

### 2.6. Curcumin Protects the Brain from the Oxidative Stress Induced by Cuprizone Administration in Mice

Brain analysis revealed that mice exposed to cuprizone for 8 weeks had a significantly increased pro-oxidant MDA level compared to the control group (*p* < 0.001) (Table 1). Moreover, cuprizone administration also significantly reduced the brain antioxidant enzymes, including SOD (*p* < 0.001), CAT (*p* < 0.001) and GSH (*p* < 0.001), compared to the CTR- group. Interestingly, treatment with curcumin significantly reduced the pro-oxidant MDA (*p* < 0.001) level and increased the activity of the antioxidant enzymes, including SOD and CAT, as well as the level of GSH (*p* < 0.001) in the mice’s brains compared to those in the CPZ and SR groups (Table 1). 

### 2.7. Curcumin Decreases Demyelination in the Corpus Callosum of Cuprizone-Mice

To confirm the demyelinating effect of cuprizone, brain sections of each group were procured at the end of 8 weeks of exposure to this neurotoxic agent and were stained with LFB, which stains with lipid-rich myelin blue, and with an antibody against MBP that binds myelin protein. Since the susceptibility to cuprizone exposure is variable in the brain, we focused on the corpus callosum region because it is one of the most studied regions in this model of demyelination (Figure 7A). As expected, the myelin was reduced in CPZ animals, as shown by LFB staining, in which the intensity of myelination was significantly decreased (Figure 7B,C).

Figure 7B indicates that in CTR- mice, the myelin sheaths showed a normal, multi-layered and compact structure. However, 8 weeks of cuprizone intake caused disorganization in the myelin composition with a significantly lower intensity compared to CTR- mice, demonstrating that cuprizone alters the lipid and protein structures of myelin. Interestingly, in the group receiving Cur after cuprizone intake, the degree of myelination was significantly increased, reaching a similar level to the CTR- group. The amount of myelin, however, remained lower in SR group compared to CUR mice (Figure 7B,C).

Furthermore, MBP expression was assessed by immunohistochemistry in the corpus callosum of all groups of mice. MBP accounts for about 30% of the total myelin proteins and is crucial for the integrity of the myelin sheath in the CNS [26]. As shown in Figure 8A, anti-MBP staining in the corpus callosum of the CPZ group was clearly reduced compared to the controls. In contrast, there was stronger anti-MBP staining in the same region for Cur-treated mice. A quantitative analysis revealed that the mean anti-MBP staining in the Cur-treated mice was significantly higher than those in the CPZ and SR groups (*p* < 0.05), but remained lower than that in the control mice (Figure 7B).

## 3. Discussion

Multiple sclerosis (MS) is a complex disease of the CNS, associated with inflammation and demyelination. Current approved therapies against MS reduce relapse rates, but they do not promote myelin repair nor do they prevent or reverse disease progression. Since we showed that curcumin treatment promotes neuroprotection in animal models [23,24], we considered that curcumin treatment may promote neuroprotection in MS animal models, which could be regulated by several mechanisms. In fact, in the present study, curcumin demonstrated a dual mechanism of action: (i) an immunomodulatory effect, through an anti-inflammatory effect in reactive astrocytes and the inhibition of immune cells’ infiltration into the spinal cord, and (ii) an ameliorating effect of myelination through the elevation of MBP expression in the brain. 

In the in vitro study, we assessed the effect of curcumin on an astrocytes cell line stimulated by LPS. Astrocytes are the most abundant type of glial cell within the mammalian brain, working as neuroprotectors against brain damage [10]. In addition, astrocytes play multiple crucial roles in the progression of MS injuries, including the recruitment of lymphocytes, stimulation of lesion recovery and confinement of inflammation [10,27]. To mimic the neuroinflammation in vitro, we incubated the murine astrocyte cell line, C8-D1A, with LPS (1 μg/mL) for 18 h, which led to a dramatic stimulation of the release of Th1/Th17 cytokines contributing to the pathogenesis of MS. Elyaman et al. reported that proinflammatory cytokines, including TNFα, and interleukins such as IL-17, IL-22 and IL-23 play crucial roles in MS development [28]. These cytokines stimulate the expression of IL-17 on microglia, astrocytes and macrophages, in an autocrine manner, leading to Th17 differentiation from naïve CD4+ T cells [6,29]. As such, Th17 cells are the key immunological player in the pathophysiological development of MS. 

On the other hand, pretreatment with curcumin at 10 μM reduced LPS-induced inflammation via the reduction of TNFα, IL-2, IL-6, IFNγ and IL-17A production. In addition, curcumin stimulated the production of IL-10, an endogenous anti-inflammatory cytokine. Our results are in accordance with those of Zhang et al., who demonstrated that curcumin inhibits LPS-induced inflammatory cytokine expression in astrocytoma cells [30] and C6 astroglial cells [31]. Curcumin also reduced the release of various inflammatory mediators (IL-1β, IL-6, TNF-α and MCP-1) in a transwell co-culture of neurons and non-neuronal cells like microglia [32], and in both primary astrocytes and primary microglia [33]. These findings suggest that the pharmacological activity of curcumin in MS models is mediated by the effective regulation of both pro- and anti-inflammatory mediators.

In addition, we found that treatment with curcumin at 10 µM stimulated the cell proliferation of C8-D1A cells. On the other hand, curcumin at 50 µM induced morphological changes and high alteration of cells when compared to the NT group and those treated with 10 µM of curcumin, who maintained their integrity as well as their interconnections. These findings support the notion that curcumin at 10 µM could promote neurotrophic factor production and neurogenesis [34].

In preclinical models of MS, we noted a dual mechanism of action that led to a significant reduction in disease severity and enhancement of myelination.

First, this study aimed to develop animal models of MS. C57BL/6 female mice are the most commonly used strain for this model as they respond perfectly to (i) immunization against myelin oligodendrocyte glycoproteins (MOG), and (ii) neurotoxicity induced by cuprizone administration. The EAE model was used first as it recapitulates several characteristic features of human MS, including chronic autoimmune inflammation, CNS demyelination and paralysis [35], in which T cells play a critical role in the induction of EAE [36]. 

Concerning EAE model development, the immunogenic epitope MOG_35–55_ is suspended in complete Freund’s adjuvant (CFA) before immunization and pertussis toxin (PTX). It is reported that PTX is an immune adjuvant utilized to effectively promote an inflammatory response in animals, as well as being involved in the pathogenesis of EAE [37]. The administration of PTX induced a pro-inflammatory cascade of IL-6, TGF-β and IL-17 in the CNS, which have been essential for the development of EAE [38]. 

In this study, mice behaviors were evaluated daily using a neurological scoring system for 45 days. The first EAE signs appeared 12 days after disease induction, consisting of weight loss and deceased activity. The maximal degree of paralysis, indicating an acute phase, was reached on day 21, when the average neurological score attained was 4.29. 

In EAE, the spinal cord is the first region affected by encephalitogenic impairments, such as inflammation and demyelination. In our study, histological analysis of the spinal cords of mice showed that EAE animals had inflammatory foci and demyelination. Furthermore, immunohistochemistry analysis showed a higher expression of LTCD3+ and LTCD4+ compared to control mice. In fact, the T-cell-mediated immune response plays a role in the pathogenesis of MS, which is recognized as a CD4+ T-cell-mediated autoimmune disease [6,39]. Once into the CNS, activated myelin specific CD4+ T cells secrete cytokines and chemokines and initiate the inflammation cascade [40]. In the early stages of EAE, the CD4+ T cells play a regulatory role, whereas CD8+ T cells seem to take over this role at the late stages [41], which is in accordance with our results.

As shown in the cellular part of this study, curcumin presents an anti-inflammatory effect via the regulation of the Th1/Th2/Th17 response. Some researchers proved that curcumin can efficiently infiltrate the CNS of EAE mice [42]. Interestingly, we found that post-treatment with curcumin at 100 mg/kg for 15 days was effective in controlling neurological symptoms of EAE and reducing disease severity when compared to the Srec group. In our study, we showed that neurological abnormalities persisted after 45 days post-MOG peptide injection. The results corroborate with those of the team of Bannerman, who showed that mice immunized with a MOG peptide in CFA show prolonged neurological disability until 98 days; however, they noted that the expression of hypophosphorylated neurofilament returned to normal values [43]. 

In addition, we showed that neuroinflammation persisted after 45 days post-MOG peptide injection. However, Constantinescu and collaborators propounded that inflammation was attenuated after the peak of the disease in the EAE model induced by MOG immunization [44]. 

Histological examination showed that curcumin attenuated neuroinflammation, clearly shown by lower inflammatory cell infiltration in the spinal cords of treated mice. This may be the consequence of the enhanced remyelination, clearly shown by the myelinated areas. Curcumin reduced the expression of LTCD4+ cells in EAE mice and induced the relocalization of these cells via the inhibition of their migration across the BBB [42]. 

In contrast to other models, cuprizone is more suited to the study of demyelination/remyelination [45,46,47]. In this model, animals are exposed to cuprizone, which induces oligodendrocytosis, resulting in demyelination associated with micro-/astro-glial activation [46,47,48]. Besides this, cuprizone increases oxidative stress and neuroinflammation [49]. Therefore, we opted for an 8-week cuprizone administration period to achieve peak demyelination as well as to mimic a chronic course of demyelination. It is not possible to evaluate the state of demyelination/remyelination without sacrificing the animals, though behavioral experiments could measure disease burden in vivo. 

The CPZ group showed severe motor dysfunction and behavioral disorder symptoms. These observations are similar to those reported by Peón et al. [50] and Sospedra et al. [51], who highlighted that cuprizone induced motor dysfunction and schizophrenia-like activity. In the present study, we found that curcumin could improve behavioral abnormalities as it caused an increase in the locomotor activity, motor coordination and memory when compared to those in the cuprizone-treated groups (CPZ and SR). 

On the other hand, cuprizone increased brain oxidative stress, as demonstrated by higher levels of MDA and lower activity of SOD and lower catalase and GSH levels. Indeed, cuprizone alters mitochondrial function in the CNS of mice and increases the generation and releases of reactive oxygen species (ROS), which causes oxidative stress [52]. ROS also directly induce oligodendrocyte death [53]. Conversely, curcumin administration at a dose of 100 mg/kg attenuated brain oxidative stress when compared to the CPZ and SR groups. In support of these findings, it has been shown that curcumin alleviates oxidative injury of the nervous system in models of Alzheimer’s disease [54], epilepsy [55] and Parkinson’s disease [56].

In addition, we found that cuprizone ingestion led to significant demyelination and decreased expression of MBP protein in the corpus callosum of mice. The corpus callosum connects the left and right cerebral hemispheres and is the largest white matter structure in the brain. Several studies on human diseases revealed corpus callosum abnormalities in the form of demyelination and decreased expression of myelin-related proteins, which are closely related to the pathogenesis of MS [57,58]. 

Treatment with curcumin increased remyelination and MBP expression in the corpus callosum of CPZ-treated mice. To our knowledge, this is the first study reporting the effect of curcumin on corpus callosum demyelination. However, several available studies, mostly on the peripheral nervous system, have indicated that curcumin accelerates the repair of sciatic nerve injury in rats by protecting myelinating Schwann cells [59,60].

Finally, this study demonstrated the mechanisms through which curcumin could provide multiple therapeutic effects in MS disease. Curcumin treatment significantly modulated inflammatory cytokines in stimulated astrocytes and decreased the clinical severity and inflammation of EAE mice. Data from an in vivo cuprizone-induced model of demyelination indicated that the observed efficacy of this biomolecule resulted directly from enhanced remyelination via the increased expression of MBP protein. Our data demonstrated, for the first time, the novel potential of curcumin as a neuro-immunomodulator and myelinating and neural repair agent, especially in the progressive stage, which is a more challenging goal to meet and thus holds great relevance for MS treatment.

## 4. Materials and Methods

### 4.1. Astrocyte Culture and Treatment

Mouse C8-D1A astrocyte cells (Astrocyte type I clone from C57/BL6 strains) were purchased from the American Type Culture Collection (ATCC^®^ CRL-2541™). C8-D1A astrocytes were plated at a density of 100,000 cells/cm^2^; they were cultured in Dulbecco’s modified Eagle medium (DMEM) (Gibco™, Sigma, St. Louis, MO, USA) supplemented with 10% fetal bovine serum and 1% penicillin/streptomycin. C8-D1A cells were incubated at 37 °C in a humidified 5% CO_2_–95% air atmosphere.

#### 4.1.1. MTT Cytotoxicity Determination

Cell viability was determined by MTT (3-[4,5-dimethylthiazol-2-yl] 2,5-iphenyltetrazolium bromide) assay according to the manufacturer’s instructions. In brief, 4 × 10^3^ C8-D1A cells were seeded overnight into 96-well plates containing proliferation medium; then, different concentrations of Cur (10, 20 and 50 μM) were added. After 2 days, 10 μL of 5 mg/mL MTT was added to each well, and cells were further incubated at 37 °C for 4 h. Following that, 100 μL of dimethylsulfoxide (DMSO) was added to each well and incubated for 10 min. Absorbance was detected at 460 nm using an ELISA plate reader.

#### 4.1.2. Determining Inflammatory Marker Concentrations Using the Multiplex Technique

C8-D1A cells were pre-incubated with Cur at 10 μM concentration for 2 h and then stimulated with LPS (1 mg/kg) for 18 h. The time of incubation with LPS was selected according to the study by Chanput and collaborators, who showed the ability of C8-D1A to reach a plateau for cytokine secretion after 18 h of LPS stimulation [59]. After that, centrifugation at 10,000× *g* (4 °C) was done to separate secreted cytokines from cell debris. Samples were stored at −80 °C for further analysis. The cytokine levels in cell-culture supernatants were quantified by AimPlex Mouse Th1/Th2/Th17 7-Plex assay kits (IL-2, TNF- α, IL-17A, IFNγ, IL-4, IL-6 and IL-10) (CliniSciences, Nanterre, France) according to the manufacturer’s instructions. Specific antibodies in the kit had been pre-coated on magnetic microparticles embedded with fluorophores at set ratios for each unique microparticle region. Briefly, capture antibody-conjugated beads were first incubated with samples or standard controls, then with biotinylated detection antibodies and finally, with streptavidin-phycoerythrin (PE). Next, final washes with kit buffer were performed to remove unbound streptavidin-PE. The plates were then analyzed using a flow cytometer (FACS Canto II, Becton Dickinson). Each sample was analyzed in triplicate. 

### 4.2. Animals and Housing

The subjects in this study were 8-week-old female mice, C57BL/6, obtained from the Pasteur Institute of Tunis (Tunisia). Animals were randomly assigned to standard cages, with two animals per cage, and kept in standard housing conditions with a light/dark cycle of 12 h in a temperature-controlled environment (22 °C, 50–60% humidity) and free access to food and water. 

Experiments were carried out in accordance with the guidance of the Internal Institutional Committee of the Pasteur Institute of Tunis (No. 2022/3/I). All efforts were made to minimize the number of animals used. 

### 4.3. Encephalomyelitis Autoimmune Experimental Model and Experimental Design

A total of 18 mice were immunized with an emulsion of the antigen (myelin oligodendrocyte glycoprotein (MOG_35–55_) and complete Freund’s adjuvant (CFA) supplemented with 1 mg/mL of heat-killed *Mycobacterium tuberculosis* and pertussis toxin, which operate by causing the formation of myelin autoreactive T cells, which mediate inflammatory disease of the CNS. Briefly, animals were anesthetized, and 100 μL of MOG_35–55_ emulsified in Freund’s adjuvant were injected subcutaneously in the upper and lower back, at a total volume of 200 μL per animal. At 2 and 24 h after MOG administration, 80 ng of pertussis toxin was administered via an intraperitoneal route to each animal to break down the BBB. This group of mice was called EAE. In parallel, control mice (*n* = 6) named CTR- received PBS via an intraperitoneal route. 

Animals were weighed and scored daily as mentioned in the literature [16]. Neurological scores were used for the assessment of motor deficits associated with EAE after disease induction based on the following scoring: 0 no disease, 1 limp tail, 2 hindlimb weakness, 3 complete hindlimb paralysis, 4 hindlimb + forelimb paralysis, 5 dead.

After 21 days, the EAE animals were divided into three groups (Figure 9): -the EAE group (*n* = 6) was euthanized at the same time as the CTR- group;-the second group (*n* = 6) received, via gavage, a daily dose of curcumin (100 mg/kg) dissolved in corn oil for an additional 15 days. The animals were weighed and scored daily;-the third group (*n* = 6) named SRec did not receive anything; this group was included to determine the spontaneous recovery rate. The animals were weighed and scored daily.

At day 45, animals from the second and third groups were sacrificed using an overdose of anesthesia. Vertebral columns were dissected, then fixed in paraformaldehyde. Spinal cords from all groups were collected and used for histological and immunohistochemical analysis.

Negative group (CTR-), cuprizone-mice (CPZ) group, post-treated with curcumin (CUR) group and spontaneous remyelination mice (SR) group.

#### 4.3.1. Histological Examination of Spinal Cord

Histological examination aimed to assess the degree of neuroinflammation in EAE and treated mice. The spinal cord was removed quickly from each group. A portion of the spinal cord sample was fixed in 10% neutral buffered formalin and then cut into 4 µm-thick sections. Tissues were stained with hematoxylin and eosin (HE) and luxol fast blue (LFB) (Sigma, St. Louis, MO, USA). After that, tissues were observed under an optical microscope (Olympus BX51 microscope with an Olympus DP12 digital camera). 

#### 4.3.2. Immunohistochemical Analysis of Inflammatory Cells in Spinal Cord

For immunohistochemical staining, spinal cord sections were subjected to antigen retrieval using a microwave in 0.01 mol/L citrate solution for 15 min, then incubated with rabbit anti-CD3, anti-CD4 and anti-CD8 antibodies (Abcam kit). The sections were then incubated with a streptavidin-biotin-peroxidase complex (SABC) kit for 30 min and stained with 3,3′-diaminobenzidine (DAB). The mounted sections were sealed with epoxy resin, and the results were observed under a light microscope (Olympus BX51 microscope with an Olympus DP12 digital camera).

### 4.4. Demyelination Experimental Model and Experimental Design

A total of 18 mice were administered a daily dose of cuprizone (bis(cyclohexanone)oxaldihydrazone) of 300 mg/kg (Sigma-Aldrich Inc., St Louis, MO, USA), emulsified in 0.5 mL of corn oil, via oral gavage. This group was called CPZ. At the same time, negative control mice (CTR-) received 0.5 mL of corn oil. After 8 weeks, CPZ mice were divided into three groups of six animals/group as presented in the following steps (Figure 10):
Step 1:
-The first group (CPZ) (*n* = 6) contained animals that received cuprizone for 8 weeks and were sacrificed in parallel with the negative control group.
Step 2:
-The second group (CUR) (*n* = 6) consisted of mice intoxicated with cuprizone and post-treated with 100 mg/kg of curcumin via oral gavage for 15 days;-The last group was named SR (*n* = 6) and referred to the mice that underwent spontaneous remyelination after halting cuprizone administration.

#### 4.4.1. Behavioral and Motricity Studies

At the end of steps 1 and 2, behavioral studies were conducted using a battery of compartmental assays: The light/dark test is one of the most widely used tests to measure anxiety-like behavior in mice. Mice were allowed to move freely between the black and light chambers for 5 min. The number of transitions between the light and dark chambers and the duration of time spent in each chamber are indicators of bright-space anxiety in mice [61];The object recognition test (NORT) is a commonly used behavioral assay for the investigation of various aspects of learning and memory in mice. During both the familiarization and the test phases, objects were in opposite and symmetrical corners of the arena. Normal mice spend more time exploring the novel object during the test phase [62];

The pole test evaluates simple motor function [63]. Animals are placed on a 50–55 cm vertical pole with a diameter of 1 mm. Scoring starts when the animal initiates a turning movement. The time spent on the rod (Trod) and the period required to descend to the ground (Tdescend) are recorded. Three tests are performed per animal and the average of Trod and Tdescend is used for data analysis;

3.The inverted screen test was used to assess the muscle strength of all four limbs [64]. The inverted screen is a 43 cm square of wire mesh consisting of 12-mm squares of 1 mm-diameter wire. The time was recorded when the mouse fell off or was removed when the set time of 60 s was reached. The inverted screen test is scored as follows: falling between 0 and 10 s = 1, 11 and 25 s = 2 and 26 and 60 s = 3, or reaching 60 s = 4.

#### 4.4.2. Biochemical Parameters’ Determination

After behavioral studies, the animals were anesthetized; then, the brains were removed and homogenized in cold phosphate buffer saline (PBS), containing a mixture of protease inhibitors, before centrifugation at 15,000× *g* for 15 min. The resulting supernatant was collected and kept at −80 °C for further analysis. Lipid peroxidation was detected by determining the malondialdehyde (MDA) production in the brain tissue following the protocol of Buege et al. [65]. Tissue superoxide dismutase (SOD) activity was assessed according to Misra and colleagues’ method [66]. Tissue catalase (CAT) activity was assayed as described by Aebi et al. [67]. The total glutathione (GSH) content in tissue was measured by the method of Tietzi et al. [68] using dithionitrobenzene and analyzed spectrophotometrically at 412 nm.

#### 4.4.3. Histopathological Examination

For histological assessment, brains were fixed in 10% paraformaldehyde solution. The degree of demyelination in the corpus callosum was assessed by Luxol Fast Blue (LFB) staining (Sigma, St. Louis, MO, USA). Briefly, paraffin-embedded samples were put on slides, deparaffinized, rehydrated by reducing concentrations of ethanol and placed in the LFB solution (0.01%) at 60 °C. To distinguish between white and gray matter, tissues were differentiated using a lithium carbonate solution (0.05%, Merck, Darmstadt, Germany). The stained tissues were observed and captured using a light microscope (Olympus BX51 microscope with an Olympus DP12 digital camera). The clear-white areas of demyelination relative to the blue normal tissue were measured and quantified using a total of 10 random sections per mouse. 

#### 4.4.4. Immunohistochemistry for MBP Protein 

For immunostaining, in brief, prepared brain sections were deparaffinized and rehydrated. To prevent non-specific binding, the samples were treated with 0.1% bovine serum albumin (BSA) (Sigma, St. Louis, MO, USA) in 0.1% Triton X-100/PBS for 1 h. Afterward, the sections were incubated overnight with the mouse anti-MBP primary antibody (1:4000, Abcam, Cambridge, UK). The sections were then exposed to the appropriate horseradish peroxidase (HRP) conjugated secondary antibody. The IHC results were quantified by measuring the immunostained areas using ImageJ software. Data were presented for the percentage of reactive areas and mean staining intensity. 

### 4.5. Image Analysis

Images were acquired on an Olympus BX51 microscope at 10× and 20× magnifications with an Olympus DP21 color camera. All microscope and camera settings (such as light level, exposure, gain) were identical for all acquired images. The software used to take image was Image Pro Plus (v6.2). All images were analyzed using ImageJ software (publicly available version). A circular region of interest (ROI) of the standard size was manually placed in the center of the lesion on microphotographs of anatomically matched LFB- and MBP-stained sections. ROIs were placed approximately in the center of the corpus callosum. Myelin densities on LFB-stained photographs were quantified based on the measurement of the mean intensity of the red channel (IR) in a ROI region [69,70]. Images were analyzed with ImageJ software using the “threshold” function and the same procedures were applied to all images. Images were converted to 8-bit and then demyelination areas were identified using threshold algorithms. Specifically, the Otsu threshold algorithm was applied to identify positively marked demyelinating areas [71]. MBP expression was also quantified from stained sections using the Otsu threshold method in ImageJ. 

### 4.6. Statistical Study

In the EAE model, the results were expressed by giving the mean of the values of each group and the standard error of the mean (SEM). Data between two groups were compared using Student’s *t*-test. Comparison of the data from multiple groups against one group was performed using a one-way analysis of variance (ANOVA) followed by Tukey’s post-hoc test. A value of *p* < 0.05 was considered statistically significant.

*: comparisons between the EAE group and CTR- group 

#: comparisons between the treated group (CUR) and EAE group

In the cuprizone model, the results were expressed by giving the mean of the values of each group and the standard error of the mean (SEM). Comparisons between groups were conducted by one-way analysis of variance (ANOVA) followed by Tukey’s post-hoc test. The CPZ and control groups were compared, as were the CPZ and CUR groups. Differences with *p* < 0.05 were considered statistically significant; * *p* < 0.05, ** *p* < 0.01, *** *p* < 0.001.

## 5. Conclusions

In summary, there is an urgent need to develop remyelination treatments to reduce the burden of demyelinating diseases such as MS. The key to developing these treatments lies in identifying safe therapeutic targets that enable myelin repair and restoration of lost functions. The beneficial effects of curcumin on the improvement of remyelination were proven in both EAE and CPZ animal models. Curcumin can enter the CNS through the BBB in both animal models and improve the pathogenic cellular and molecular mechanisms responsible for motor function decline, inflammatory infiltration, downregulation of MBP expression and demyelination. Curcumin has potential for treating MS, and researchers must continue to focus on more efficacious curcumin-based drugs or the used of curcumin-based drugs in combination with routinely used treatments.

## Figures and Tables

**Figure 1 ijms-23-08658-f001:**
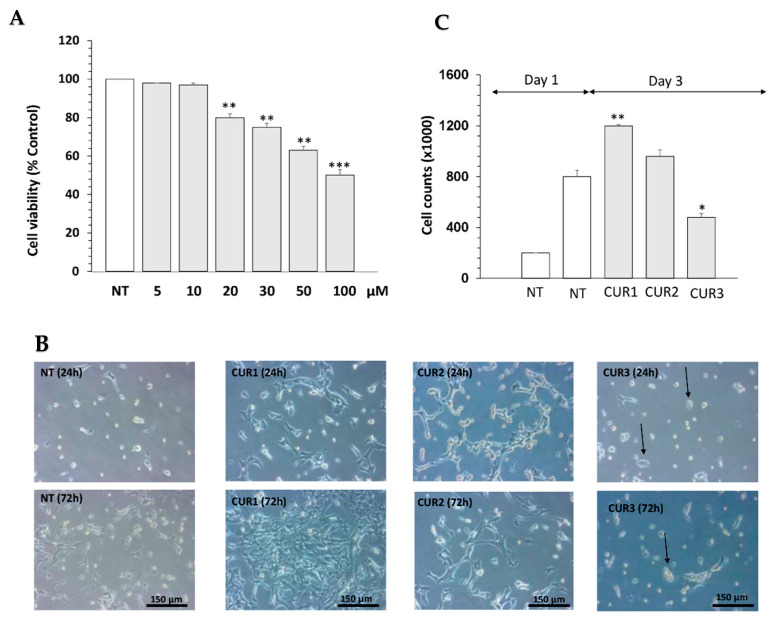
Effect of curcumin on cell viability and proliferation of astrocyte C8-D1A cells. (**A**) Cell viability of astrocyte C8-D1A cells was determined by MTT assay after 24 h of treatment. (**B**) Aspect of cells treated with different concentrations of Cur (10, 20 and 50 µM); black arrows show circular form of cell and indicate that 50 µM induced a change in the morphology of cells. (**C**) Number of cells after 72 h of incubation with Cur at 10, 20 and 50 µM. Data are shown as mean ± SEM of three independent experiments in a triplicate assay. * *p* < 0.05, ** *p* < 0.01 and *** *p* < 0.001.

**Figure 2 ijms-23-08658-f002:**
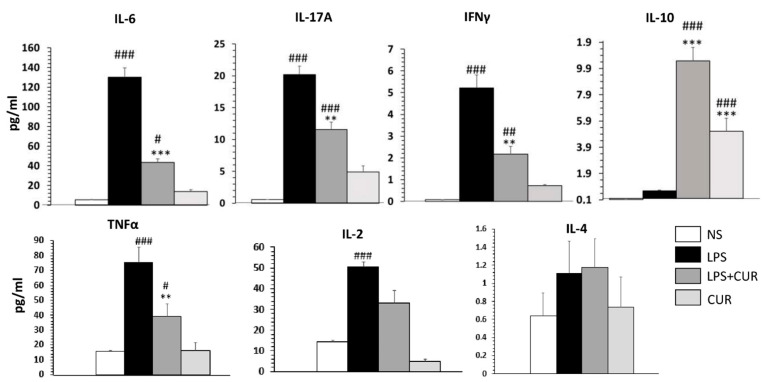
Effect of curcumin on the Th1/Th2/Th17 cytokines release by LPS-stimulated C8-D1A cells. Data are shown as mean ± SEM of three independent experiments in a triplicate assay. ^#^ *p* < 0.05, **/^##^ *p* < 0.01 and ***/^###^ *p* < 0.001.

**Figure 3 ijms-23-08658-f003:**
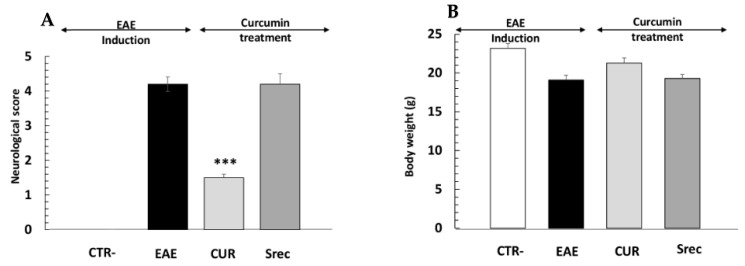
Effect of curcumin on the neurological signs of EAE. (**A**) Neurological scores were recorded daily, then analyzed, plotted and compared between groups. (**B**) Body weight of animals in experimental groups during the study. Negative group (CTR-), encephalomyelitis autoimmune experimental (EAE) group, post-treated with curcumin (CUR) group, spontaneous recovery (Srec) group. Values are given as mean ± SEM as the results of ordinary one-way ANOVA followed by Tukey’s multiple comparison test. *** *p* < 0.001.

**Figure 4 ijms-23-08658-f004:**
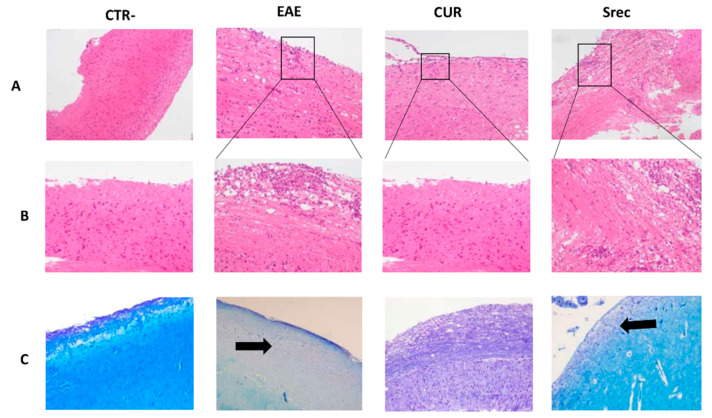
Effect of curcumin on the inflammation and demyelination in the spinal cord of EAE mouse. (**A**) Hematoxylin and eosin staining for assessment of inflammation (Obj × 10). (**B**) Boxes in Figure 4A show magnified images of inflamed region (Obj × 20). Higher inflammatory cell infiltration was observed in the EAE group when compared to CTR- animals; Cur alleviated inflammation when compared to the EAE and Srec groups (**C**). Representative micrographs prepared from the spinal cords of all groups, stained with luxol fast blue (Obj × 10). Arrows show magnified images of demyelinated areas; 10 sections from each group were scored.

**Figure 5 ijms-23-08658-f005:**
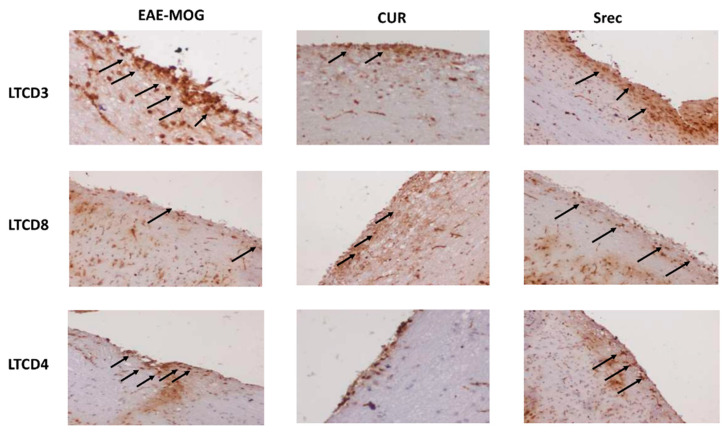
Effect of curcumin on the expression of immune cells in the spinal cord of EAE mice. Immune cell infiltration was studied using immunostaining of mice spinal cords with anti-CD3, CD4 and CD8 antibodies (Obj × 20). Arrows show an increase in the expression of LTCD3 and LTCD4. Cur treatment decreased the expression of inflammatory markers in the EAE lesions; 10 sections from each group were scored.

**Figure 6 ijms-23-08658-f006:**
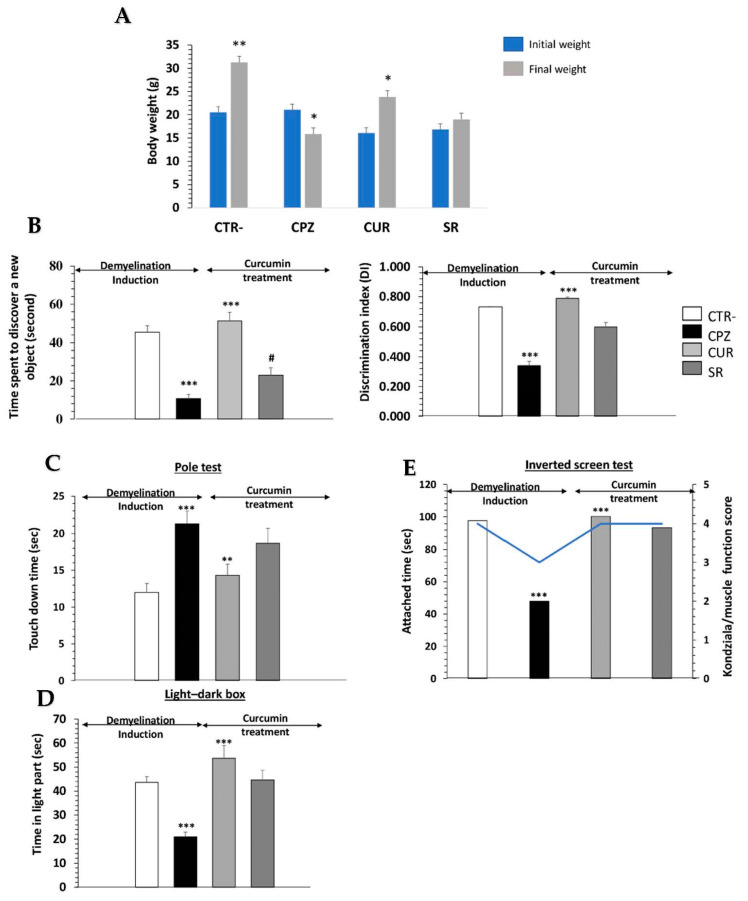
Effect of curcumin on cuprizone-induced behavioral changes in mice. (**A**) Body weights of animals in experimental groups at the beginning and end of experimentation, as well as during the study. Performance of animals in different cognitive tasks including novel object discrimination (**B**), pole (**C**), light/dark box (**D**) and inverted screen (**E**) tests. One-way ANOVA followed by Tukey’s multiple comparison test; */^#^ *p* < 0.05, ** *p* < 0.01 and *** *p* < 0.001. *** *p* < 0.001 shows significant differences between negative control and cuprizone groups. ** *p* < 0.01 and *** *p* < 0.001 show significant differences between cuprizone and curcumin-treated groups.

**Figure 7 ijms-23-08658-f007:**
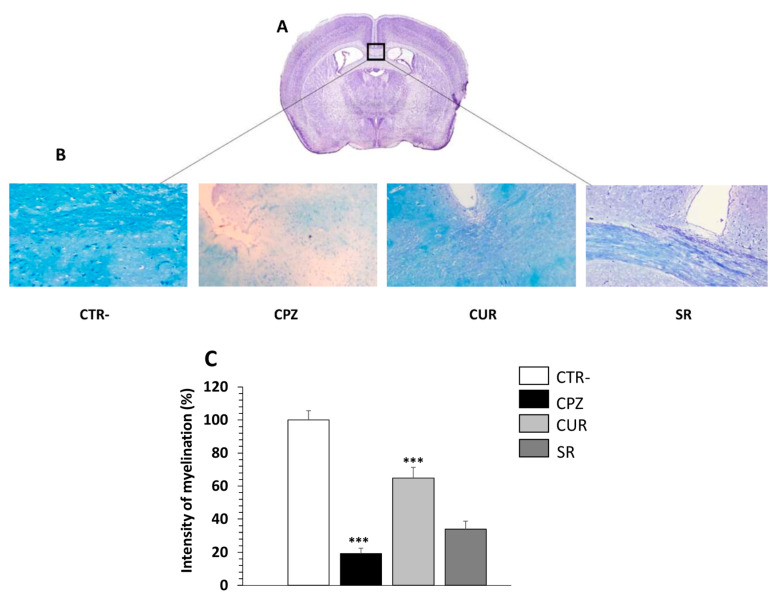
Effect of curcumin treatment on demyelination induced by cuprizone intake in the corpus callosum of mice. (**A**) Adapted from Paxinos Franklin Mouse Brain Atlas. (**B**) Representative images for LFB staining results (Obj × 20). (**C**) Evaluation of myelination index by analyzing LFB intensity. *n* = 10 brain sections from three mice, per experimental group. Negative group (CTR-), cuprizone-mice (CPZ) group, post-treated with curcumin (CUR) group and spontaneous remyelination mice (SR). Values are given as mean ± SEM, as the results of ordinary one-way ANOVA followed by Tukey’s multiple comparison test; *** *p* < 0.001.

**Figure 8 ijms-23-08658-f008:**
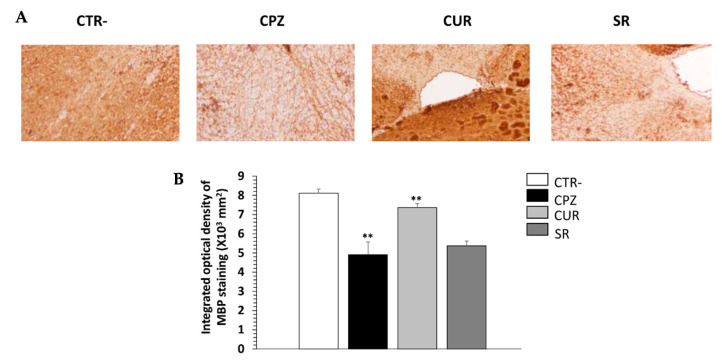
Effect of curcumin treatment on myelin content of corpus callosum during cuprizone intake. (**A**) Representative images for myelin basic protein (MBP) staining results (Obj × 20). (**B**) Quantitative analysis of MBP expression. *n* = 10 brain-sections from three mice, per experimental group. Negative group (CTR-), cuprizone-mice (CPZ) group, post-treated with curcumin (CUR) group and spontaneous remyelination mice (SR). Values are given as mean ± SEM as the results of ordinary one-way ANOVA followed by Tukey’s multiple comparison test; ** *p* < 0.01.

**Figure 9 ijms-23-08658-f009:**
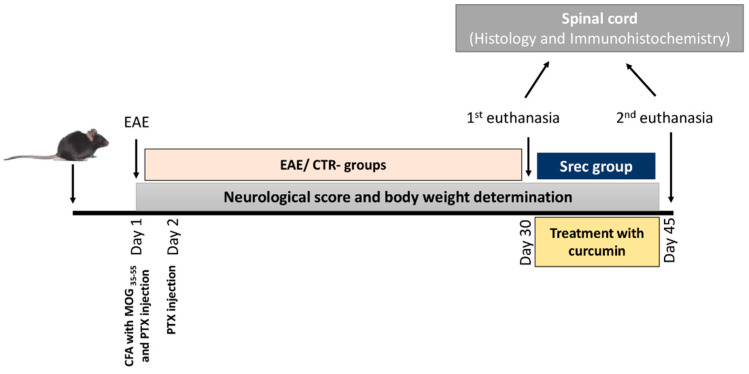
Schematic description of demyelination experimental design.

**Figure 10 ijms-23-08658-f010:**
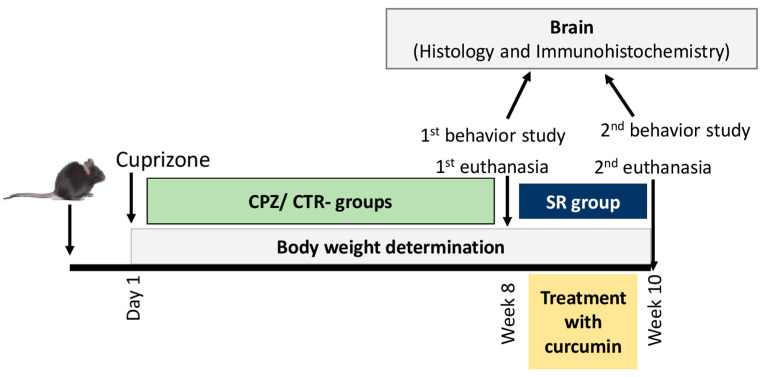
Schematic description of encephalomyelitis auto-immune experimental (EAE) design. Negative group (CTR-), EAE-mice (EAE) group, post-treated with curcumin (CUR) group and spontaneous recovery mice (Srec) group.

**Table 1 ijms-23-08658-t001:** Effect of curcumin on the oxidative stress markers induced by cuprizone administration in mice.

	MDA (nM/mg of Protein)	SOD (U/mg of Protein)	CAT (U/mg of Protein)	GSH (nM/mg of Protein)
CTR-	0.52 ± 0.23	6.24 ± 0.56	100.23 ± 6.08	9.12 ± 0.29
CPZ	1.62 ± 0.52 ***	2.54 ± 0.62 ***	54.69 ± 10.36 ***	3.69 ± 0.19 ***
CUR	0.81 ± 0.32 **	7.23 ± 0.87 **	116.23 ± 6.08 ***	8.14 ± 0.301 ***
SR	1.1 ± 0.29	4.57 ± 0.58	70.89 ± 8.52	5.23 ± 0.42

Each value represents the mean ± SEM of six animals. MDA: malondialdehyde; SOD: superoxide dismutase; CAT: catalase; GSH: glutathione dismutase. One-way ANOVA followed by Tukey’s multiple comparison test. *** *p* < 0.001 shows significant differences between negative control and cuprizone groups; ** *p* < 0.01 and *** *p* < 0.001 show significant differences between cuprizone and curcumin-treated groups.

## Data Availability

Not applicable.

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
