# Peer review of "Dual Mechanism of Action of Curcumin in Experimental Models of Multiple Sclerosis"

_ijms, 2022, doi:10.3390/ijms23158658_

Round 1
Reviewer 1 Report
The manuscript by ELBini-Dhouib aimed to assess curcumin as a potential bioactive substance for MS treatment. The authors use two widespread models of MS: cuprizone-induced demyelination and EAE, as well as astrocyte cell culture to assess curcumin toxicity and effect on inflammation. The authors have done a great job to prove the potential benefits of using curcumin in MS. However, the manuscript contains a large number of ambiguities that should be clarify and corrected.
Overall comments:
- The authors evaluate astrocyte cell culture but did not evaluate astrocytes in the cuprizone and EAE models.
- Since there is no clear description or scheme of treatments for ALL groups in the Methods section, it is difficult to assess the correctness of the experimental design. Was there a parallel control for curcumin groups? For the EAE model, as follows from Figure 3B, parallel control is not shown. In this case, it cannot be argued that the improvement is caused by curcumin and is not spontaneous. A decrease of inflammation and remyelination may occur with EAE (https://www.ncbi.nlm.nih.gov/pmc/articles/PMC3229753/).
Introduction.
- Lines 82-88 – These lines already include conclusion before we see the “Methods” and “Results” sections. In my opinion, the "Introduction" section can outline the goals and objectives of the study but cannot contain a conclusion.
Methods.
- The Methods section is poorly structured. Usually there is a separate point "Animals and housing", which describes not only the sample size, line, origin, age and gender of animals (this information is in the manuscript), but also describes the housing (light/dark schedule, temperature, feeding). This point also should contain an important information concerning the approval of the local ethical committee. The number of animals in each subgroup are also should be specified.
- The timeline of the experiment is not clear. Were the animals from all groups sacrificed at the same time or at different time points? I advise the authors to add the scheme showing all treatments, behavioral tests and other time points in the Method instead Results section.
- Histological and immunohistochemical staining, as well as image processing are usually described in separate sections.
- The model and manufacturer of the light microscope, objective parameters (or zoom) as well as the software for photographing are not specified.
- Image processing is also poorly described. How were regions of interest selected for image processing? What methods were used to quantify LFB and MBP? Was the red channel or grayscale used to quantify the LFB images? Usually, the red channel is used to quantify LFB optical density (https://www.nature.com/articles/srep46686, https://elifesciences.org/articles/61523) . Was the Otsu threshold or another method used to evaluate myelinated areas? https://www.ncbi.nlm.nih.gov/pmc/articles/PMC6594192/
- The authors wrote:” LFB images were quantified as previously described” but a reference is absent. Described where?
- The title Statistical looks strange. Maybe Statistical analysis? The point is written very confusingly and carelessly.
- The authors wrote (lines 522-523, 530-531): In the EAE model, the results are expressed by giving the mean of the values of each group and the standard deviation (SEM). Standard deviation (SD) or standard error of mean (SEM)?
- The authors wrote (lines 523-524): Analyses of variance, then an ANOVA test is used for… ANOVA - this is the analysis of variance, the sentence is incorrect. Perhaps the authors mean post-hoc tests, but this should be written correctly. The type of test used must also be indicated (Bonferroni, Tukey or other). Has the Greenhouse-Geiser correction been used for multiple comparisons?
- Lines 525-526 are out of place.
Results.
- Lines 133,134,138 etc: “Clinical scores” and “clinical severity” are not applicable for mice, only for patients. Use “neurological scores” instead.
- For the cuprizone model, were behavioral tests performed twice for groups with euthanasia at a later point? Repeated presentation of the test significantly affects the results (in particular, the object recognition test). The test results at what time point are shown in Figure 5?
- Line 249: Curcumin decreases demyelination in the corpus of cuprizone- mice. Corpus callosum?
- Figure 6. In some microphotographs, the corpus callosum is not guessed at all - LFB and MBP for the curcumin group, MBP for the curcumin and SR groups. It is similar to a cortex in structure. Despite demyelination, the fiber structure in the CC is clearly seen in this model. Ideally, the photo should show larger part of the corpus callosum.
- Part of the micrographs with LBF staining in fig.5-6 have a strange lilac color. This may have affected the quantification.
Author Response
Comments of Reviewer #1
The manuscript by ELBini-Dhouib aimed to assess curcumin as a potential bioactive substance for MS treatment. The authors use two widespread models of MS: cuprizone-induced demyelination and EAE, as well as astrocyte cell culture to assess curcumin toxicity and effect on inflammation. The authors have done a great job to prove the potential benefits of using curcumin in MS. However, the manuscript contains a large number of ambiguities that should be clarify and corrected.
First of all, we would like to thank Reviewer #1 for their valuable time and expertise in the review of our work. The corrections/additions/clarifications made, as suggested by Reviewer #1 and the 2nd Reviewer we feel has resulted in a particularly strengthened submission for this research paper.
Overall comments:
Overall comment 1: The authors evaluate astrocyte cell culture but did not evaluate astrocytes in the cuprizone and EAE models.
Response 1: Authors thank the reviewer for this comment. Indeed, confirming invitro results by an invivo method seems to be ideal in this study. However, the evaluation of astrocyte function in animal models needs to culture a primary astrocyte, which is difficult to do in our cell culture room dedicated only to cell lines studies. In addition, following the recommendations of the local ethics committee, we used a limited number of animals that were involved in histological and biochemical assessments.
Overall comment 2: Since there is no clear description or scheme of treatments for ALL groups in the Methods section, it is difficult to assess the correctness of the experimental design. Was there a parallel control for curcumin groups? For the EAE model, as follows from Figure 3B, parallel control is not shown. In this case, it cannot be argued that the improvement is caused by curcumin and is not spontaneous. A decrease of inflammation and remyelination may occur with EAE (https://www.ncbi.nlm.nih.gov/pmc/articles/PMC3229753/).
Response 2: As recommended by Reviewer #1, the spontaneous recovery mice group (Srec) (parallel group) was added in the results and this part was re-written in the method section of the manuscript; figures of models were also changed. The results of spontaneous recovery group were discussed based on the review of Constantinescu et al. (https://www.ncbi.nlm.nih.gov/pmc/articles/PMC3229753/) and of Bannerman et al.(https://academic.oup.com/brain/article/128/8/1877/481392?login=true)
Introduction.
Comment: Lines 82-88 – These lines already include conclusion before we see the “Methods” and “Results” sections. In my opinion, the "Introduction" section can outline the goals and objectives of the study but cannot contain a conclusion.
Response: We agree with the reviewer. This part was re-written.
Methods.
Comment: The Methods section is poorly structured. Usually there is a separate point "Animals and housing", which describes not only the sample size, line, origin, age and gender of animals (this information is in the manuscript), but also describes the housing (light/dark schedule, temperature, feeding). This point also should contain an important information concerning the approval of the local ethical committee. The number of animals in each subgroup are also should be specified.
Response: The material and methods section was re-written and structured as recommended by the reviewer #1
Comment: The timeline of the experiment is not clear. Were the animals from all groups sacrificed at the same time or at different time points? I advise the authors to add the scheme showing all treatments, behavioral tests and other time points in the Method instead Results section.
Response:1/ As requested by the reviewer, the timeline was corrected to be clearer. Indeed, animals from negative control and 1st sub-group of MS model were sacrificed in the time 1. The rest of MS model were divided into two other groups (treated group with curcumin and spontaneous recovery group) and continued the protocol to the 2nd time of sacrifice. For each time, behavior studies were performed.
2/ As advised by the reviewer, the description of all studies was added in the method section.
Comment: Histological and immunohistochemical staining, as well as image processing are usually described in separate sections.
Response: Separate sections were added in the manuscript.
Comment: The model and manufacturer of the light microscope, objective parameters (or zoom) as well as the software for photographing are not specified.
Response: Images were acquired on an OLYMPUS BX51 microscope at 10x and 20x magnification with an OLYMPUS DP21 color camera. The software used to take image was Image Pro Plus (v6.2). Image analysis part was now added in the “method” section.
Comment: Image processing is also poorly described. How were regions of interest selected for image processing? What methods were used to quantify LFB and MBP? Was the red channel or grayscale used to quantify the LFB images? Usually, the red channel is used to quantify LFB optical density (https://www.nature.com/articles/srep46686, https://elifesciences.org/articles/61523). Was the Otsu threshold or another method used to evaluate myelinated areas? https://www.ncbi.nlm.nih.gov/pmc/articles/PMC6594192/
Response: Authors thank reviewer #1 for this comment. Histological and immunohistochemical slides were read by our anatomopathologist collaborators at the anatomopathology department of Salah Azaeiz Institute (Tunisia). Indeed, all images were analyzed using the public version of ImageJ software. A circular region of interests (ROI) of the standard size was manually placed in the center of the lesion on microphotographs of anatomically matched LFB and MBP-stained sections. Myelin densities on LFB stained photographs were measured based on of the mean intensity of the red channel (IR) in a ROI region. Images were analyzed with ImageJ software using the “threshold” function and the same procedures were applied to all images. Images were converted to 8-bit and then demyelination areas were identified using threshold algorithms.
The description of image analysis part was added in the “method” section of the actual version
Comment: The authors wrote:” LFB images were quantified as previously described” but a reference is absent. Described where?
Response: This reference was added : Xie M, Tobin JE, Budde MD, Chen CI, Trinkaus K, Cross AH, McDaniel DP, Song SK, Armstrong RC. Rostrocaudal analysis of corpus callosum demyelination and axon damage across disease stages refines diffusion tensor imaging correlations with pathological features. J Neuropathol Exp Neurol. 2010 ;69(7):704-16. doi: 10.1097/NEN.0b013e3181e3de90.
Comment: The title Statistical looks strange. Maybe Statistical analysis? The point is written very confusingly and carelessly.
Response: The title was changed as required by the reviewer.
Comment: The authors wrote (lines 522-523, 530-531): In the EAE model, the results are expressed by giving the mean of the values of each group and the standard deviation (SEM). Standard deviation (SD) or standard error of mean (SEM)?
Response: We corrected this mistake; indeed, it is standard error of mean (SEM)
Comment: The authors wrote (lines 523-524): Analyses of variance, then an ANOVA test is used for… ANOVA - this is the analysis of variance, the sentence is incorrect. Perhaps the authors mean post-hoc tests, but this should be written correctly. The type of test used must also be indicated (Bonferroni, Tukey or other). Has the Greenhouse-Geiser correction been used for multiple comparisons?
Response: Authors thank Reviewer#1 for this comment. Indeed, statistical analyses were performed by one-way ANOVA with a Tukey post hoc test.
Comment: Lines 525-526 are out of place.
Response: It was corrected.
Results:
Comment: Lines 133,134,138 etc: “Clinical scores” and “clinical severity” are not applicable for mice, only for patients. Use “neurological scores” instead.
Response: Authors thank the reviewer for this advice. We agree that the word “Clinical” used in a large number of scientific publications is not applicable for animals and “neurological” is more suitable for this item of experiment. The terms were changed in the text.
Comment: For the cuprizone model, were behavioral tests performed twice for groups with euthanasia at a later point? Repeated presentation of the test significantly affects the results (in particular, the object recognition test). The test results at what time point are shown in Figure 5?
Response: Behavioral study was performed in two-time scales, the first one after 8 weeks of cuprizone administration and the second one after two weeks of curcumin treatment. Accordingly, the legend of figures is modified, as suggested by the reviewer,
Comment: Line 249: Curcumin decreases demyelination in the corpus of cuprizone- mice. Corpus callosum?
Response: The sentence is corrected in the manuscript. Corrected sentence is “Curcumin decreases demyelination in the corpus callosum of cuprizone- mice”
Comment: Figure 6. In some microphotographs, the corpus callosum is not guessed at all - LFB and MBP for the curcumin group, MBP for the curcumin and SR groups. It is similar to a cortex in structure. Despite demyelination, the fiber structure in the CC is clearly seen in this model. Ideally, the photo should show larger part of the corpus callosum.
- Part of the micrographs with LBF staining in fig.5-6 have a strange lilac color. This may have affected the quantification.
Response: The microphotographs in figure 6 were modified as recommended by the reviewer #1. You kindly find microphotographs of demyelination in figure 7 and of those of immunohistochemistry in figure 8.
----------------------------------------------
END OF ALL COMMENTS AND RESPONSES
Dr Ines ELBini Dhouib, PhD.
Assistant Professor of Neurophysiology.
Laboratory of Biomolecules, Venoms and Theranostic Applications.
Pasteur Institute of Tunis, Tunisia.

Reviewer 2 Report
The article by Ines ELBini-Dhouib and collaborators is suitable for publication. A major revision for English is required (corrections and text editing).
Author Response
Comment: The article by Ines ELBini-Dhouib and collaborators is suitable for publication. A major revision for English is required (corrections and text editing).
We would like to thank Reviewer #2 for their valuable time and expertise in the review of our work.
Response: The revised manuscript has been once again gone over by all authors that have also critically and objectively reviewed and endorsed this paper. In addition, it was be revised by an English-speaking colleague.
----------------------------------------------
END OF ALL COMMENTS AND RESPONSES
Dr Ines ELBini Dhouib, PhD.
Assistant Professor of Neurophysiology.
Laboratory of Biomolecules, Venoms and Theranostic Applications.
Pasteur Institute of Tunis, Tunisia.
